# Differential Influence of Soluble Dietary Fibres on Intestinal and Hepatic Carbohydrate Response

**DOI:** 10.3390/nu13124278

**Published:** 2021-11-27

**Authors:** Matthew G. Pontifex, Aleena Mushtaq, Gwenaëlle Le Gall, Ildefonso Rodriguez-Ramiro, Britt Anne Blokker, Mara E. M. Hoogteijling, Matthew Ricci, Michael Pellizzon, David Vauzour, Michael Müller

**Affiliations:** 1Norwich Medical School, University of East Anglia, Norwich NR4 7TJ, UK; m.pontifex@uea.ac.uk (M.G.P.); a.mushtaq@uea.ac.uk (A.M.); g.le-gall@uea.ac.uk (G.L.G.); i.rodriguez-ramiro@uea.ac.uk (I.R.-R.); bablokker@gmail.com (B.A.B.); marahoog@xs4all.nl (M.E.M.H.); d.vauzour@uea.ac.uk (D.V.); 2Research Diets, Inc., New Brunswick, NJ 08901, USA; matthew.ricci@researchdiets.com (M.R.); michael.pellizzon@researchdiets.com (M.P.)

**Keywords:** psyllium, inulin, microbiome, metabolome, carbohydrate metabolism

## Abstract

Refined foods are commonly depleted in certain bioactive components that are abundant in ‘natural’ (plant) foods. Identification and addition of these ‘missing’ bioactives in the diet is, therefore, necessary to counteract the deleterious impact of convenience food. In this study, multiomics approaches were employed to assess the addition of the popular supplementary soluble dietary fibers inulin and psyllium, both in isolation and in combination with a refined animal feed. A 16S rRNA sequencing and ^1^H NMR metabolomic investigation revealed that, whilst inulin mediated an increase in *Bifidobacteria*, psyllium elicited a broader microbial shift, with *Parasutterella* and *Akkermansia* being increased and *Enterorhabdus* and *Odoribacter* decreased. Interestingly, the combination diet benefited from both inulin and psyllium related microbial changes. Psyllium mediated microbial changes correlated with a reduction of glucose (R −0.67, −0.73, respectively, *p* < 0.05) and type 2 diabetes associated metabolites: 3-methyl-2-oxovaleric acid (R −0.72, −0.78, respectively, *p* < 0.05), and citrulline (R −0.77, −0.71, respectively, *p* < 0.05). This was in line with intestinal and hepatic carbohydrate response (e.g., *Slc2a2*, *Slc2a5*, *Khk* and *Fbp1*) and hepatic lipogenesis (e.g., *Srebf1* and *Fasn*), which were significantly reduced under psyllium addition. Although established in the liver, the intestinal response associated with psyllium was absent in the combination diet, placing greater significance upon the established microbial, and subsequent metabolomic, shift. Our results therefore highlight the heterogeneity that exists between distinct dietary fibers in the context of carbohydrate uptake and metabolism, and supports psyllium containing combination diets, for their ability to negate the impact of a refined diet.

## 1. Introduction

Refined foods are becoming increasingly prominent in the modern diet, due to marketing, accessibility, convenience, and palatability [1]. Typical refined foods tend to be nutrient poor and calorie dense, and generally fail to provide a range of crucial dietary components, including an adequate fiber content, which can have deleterious health consequences. Although a requisite for general health and wellbeing, inadequate dietary fiber (DF) intake is now widespread; with recommended levels failing to be met by numerous developed nations [2]. Lack of DF is associated with the development of various metabolic diseases, including heart disease, stroke, cancer, and diabetes [3,4]. This may be attributable in part to DF’s role in carbohydrate uptake/metabolism [5], although further mechanistic elucidation is required.

The gut microbiota is particularly sensitive to dietary change and is frequently cited for its involvement in nutritional diseases, with dysregulation of microbial metabolites, increased intestinal barrier permeability, and endotoxemia [6] forwarded as potential mechanistic factors. DFs, like many prebiotics, have the propensity to modulate the gut microbiota, although their distinct structural diversity (e.g., inulin vs psyllium) may result in divergent microbial responses [7], ultimately affecting their capacity to regulate specific biological processes, such as carbohydrate metabolism. The physiological benefits associated with fiber types relate to chemical composition, which in turn dictates fermentability and solubility (reviewed extensively by [8]). Briefly, inulin is purportedly highly fermentable, whilst psyllium is only moderately fermentable. Fermentation in the distal small intestine and proximal colon provides energy and an array of metabolic substrates, e.g., SCFA, which are believed to promote specific microbiota alterations, leading to different fermentation patterns. For example, inulin reportedly leads to the preferential growth of the bacteria *Lactobacilli* and *Bifidobacteria*. In contrast less fermentable, intermediate soluble fibers, such as psyllium have a higher water-holding/gel-forming capacity, which may alter gastric emptying/nutrient absorption, ultimately affecting glucose and lipid absorption. Full characterization of the gut microbial response to specific DFs/DF combinations is, therefore, critical for a complete interpretation of these complex host–microbe interactions.

Given the present epidemiological, experimental, and clinical evidence supporting the benefits of a diet rich in DF [9], supplementation appears a plausible approach to mitigate the negative impact of a processed diet. This was demonstrated by two meta-analyses, in which an inverse relationship between DF intake and metabolic disease risk was established [10,11]. Despite the abundance of research exploring the influence of DFs upon disease risk and incidence [10,12,13], the mechanistic basis requires further elucidation, particularly in the context of intestinal health. Furthermore, uncertainty remains as to whether distinct DFs differ in their ability to regulate specific processes, such as carbohydrate metabolism. If so, certain DFs may offer greater protection from a processed diet than others. Therefore, further investigation of optimal DFs/DF combinations, as well as dosage, is required to identify the most suitable. In the present study, soluble DF addition (inulin, psyllium, and combination) was assessed in healthy male animals maintained on a typical refined (fiber depleted) feed, in order to explore intestinal and hepatic carbohydrate response. Multi-omics (transcriptome, microbiome, and metabolome) approaches were employed to unravel the complexity behind these interactions and determine the extent of gut microbiota/metabolome involvement.

## 2. Materials and Methods

### 2.1. Overview of Experimental Procedure

Forty male C57 BL/6J mice sourced from Charles River UK (CRUK, Margate, UK), were maintained in individually ventilated cages, within a controlled environment (21 ± 2 °C; 12-h light/dark cycle; light from 7:00 AM) and fed ad libitum on a standard chow diet (RM3-P; Special Diet Services (SDS), Essex, UK) up to the age of 10 weeks, ensuring normal development and stabilization of the microbiota [14]. After which, mice were transferred to one of four purified diets: low fat with 47.4 g cellulose/kg (LF, D12450J, cellulose, Solka Floc 200 FCC); LF with 68 g inulin/kg (LFIn, D18012101, Orafti HP inulin); LF with 68 g psyllium/kg (LFPsy, D19051001, psyllium husk powder, 95% USP); or LF with 34 g inulin/kg and 34 g psyllium/kg (LFInPsy, D19051002) (Research Diets, Inc., New Brunswick, NJ, USA) for the remaining 10-week experimental period (see Appendix A for full dietary composition). Inulin and psyllium were chosen as both are popular commercially available supplementary soluble dietary fibers (particularly for mouse feed) with reported prebiotic/health implications. Additionally, the two fibers have been predominantly studied in isolation, but are rarely compared nor assessed in combination. Upon completion of the intervention, mice were sacrificed under terminal anesthesia. Organs were rapidly collected and processed for further downstream applications, as follows. The small intestine of each mouse was collected, stripped of mesenteric fats and any non-intestinal tissue, and washed in ice cold 1X PBS. The intestinal content was gently pushed out and the remaining tissue was divided into three equal parts, approximately representing the duodenum (Part 1), the jejunum (Part 2), and the ileum (Part 3). Each section was cut open using a midline incision and intestinal mucosa was gently scraped off with the blunt edge of a glass slide and snap frozen in liquid nitrogen. Livers were excised, weighed, and consistently divided into sections for histology (left lobe) and RNA isolation (medial lobe). Samples destined for histological analysis were placed in 10% formalin for 24 h, followed by 50% ethanol until embedded, whilst the remaining tissue was snap-frozen in liquid nitrogen. Finally, the cecum was removed, weighed, and its contents were gently extracted and snap-frozen in liquid nitrogen.

### 2.2. Liver Triglyceride and Histological Analysis

Triglycerides were extracted following the Bligh and Dyer method [15] and were measured using a liquicolor colorimetric assay kit, following manufacturer’s instructions [16].

For histological analysis, samples were processed on a Leica ASP 300 overnight and embedded in Paraplast^®^ the next morning. Tissue sections were cut into 5 μm thickness using a HistoCore BIOCUT-Manual Rotary Microtome and secured on SuperFrost Plus™ Adhesion slides (Thermo Scientific, Loughborough, UK) and mounted with DPX mounting medium. Slides were then stained with haematoxylin and eosin (H&E), following the protocol from [16], and images were captured on a Olympus BX60 microscope at 10X magnification.

### 2.3. RNA Isolation and qRT-PCR

RNA isolation, cDNA synthesis, and qRT-PCR were carried out as previously described [17]. Briefly, liver and ileum samples were homogenised in Qiazol (Qiagen, Manchester, UK) at 6000 rpm in a Precellys^®^ 24 (Bertin Technologies, Aix-en-Provence, France). Following the phenol chloroform extraction, isopropanol precipitation and ethanol washes, RNA concentration was measured with a Nanodrop (Thermo Scientific, Wilmington, NC, USA). Complementary DNA (cDNA) was synthesized from 2 µg of RNA, first treated with DNase1 (Invitrogen, Loughborough, UK) and subsequently reverse transcribed using Invitrogen Oligo (dT) primers and M-MMLV reverse transcriptase (Invitrogen, Loughborough, UK). A quantitative polymerase chain reaction (qPCR) was performed using SYBR green master mix (Invitrogen, Loughborough, UK), according to the manufacturer’s instructions. Reactions were performed on QuantStudio ™ 5 Real-Time PCR System (Applied Biosystems, Thermo Fisher Scientific, Loughborough, UK). Delta CT values were calculated by subtracting the Ct value of the housekeeping gene TATA-box binding protein (*Tbp*) from the target gene Ct value. The housekeeping gene *Tbp* was used for all gene expression analysis using qPCR because of its optimal expression stability. *Tbp* has been previously reported as a constantly expressed reference gene for mouse intestine and liver tissue [18,19] Results are expressed as relative quantity scaled to the average across all samples per target gene and normalized to the reference gene, results are presented as log2 fold change. Primer sequences are given (Appendix A).

### 2.4. Microbial 16S rRNA Sequencing (Preparation and Analysis)

Microbial DNA was isolated from approximately 50 mg caecal content using the Qiagen DNA mini kit. Additional steps were added to the DNA mini kit protocol to ensure breakage of all bacterial samples. Briefly, the samples were homogenised using silica glass beads (Sigma-Aldrich, Dorset, UK) for 4 × 30 s at 6000 rpm in a Precellys^®^ 24 (Bertin Technologies, Aix-en-Provence, France) and heated to 95°C for 5 min. Additionally, samples were incubated with a lysis buffer containing 20 mg/mL lysozyme (Lysozyme from chicken egg white, Sigma-Aldrich, Dorset, UK) after which the homogenising was repeated. The lysozyme was used to help effectively capture usually difficult to lyse taxa, such as gram-positive bacteria. Consequently, DNA was isolated using the Qiagen DNA mini kit following instructions from the manufacturer. DNA quantity was assessed using a Nanodrop 2000 Spectrophotometer (Fisher Scientific, Loughborough, UK). A minimum of 50 ng of DNA was sent to Novogene (Cambridge, UK). Quality assessment was performed by agarose gel electrophoresis to detect DNA integrity, purity, fragment size and concentration. The 16S rRNA amplicon sequencing of the V3-V4 hypervariable region was performed with an Illumina NovaSeq 6000 PE250. Sequences analyses were performed by Uparse software (Uparse v7.0.1001) [20], using all the effective tags. Sequences with ≥97% similarity were assigned to the same OTUs. Representative sequence for each OTU was screened for further annotation. For each representative sequence, Mothur software was performed against the SSUrRNA database of SILVA Database. OTUs abundance information were normalised using a standard of sequence number corresponding to the sample with the least sequences. Alpha-diversity and beta diversity were assessed using Shannon H diversity index and weighted UniFrac analysis respectively. Statistical significance was determined by Kruskal–Wallis or Permutational Multivariate Analysis of Variance (PERMANOVA). Comparisons at the Phylum and Genus level were made using classical univariate analysis using Kruskal–Wallis combined with a false discovery rate (FDR) approach used to correct for multiple testing.

### 2.5. H NMR Metabolomics

Caecal metabolites were analysed and quantified by ^1^H NMR analysis. The preparation method was similar to that previously described [21]. Briefly, 20 mg of frozen caecal content were thoroughly mixed at 5000 rpm in a Precellys^®^ 24 (Bertin Technologies, Aix-en-Provence, France) in 1 mL of saline phosphate buffer [1.9 mM Na_2_HPO_4_, 8.1 mM NaH_2_PO_4_, 150 mM NaCl (MilliporeSigma, Burlington, MA, USA), and 1 mM trimethylsilyl propanoic acid [sodium 3-(trimethysilyl)-propionate-d4] in deuterated water (Goss Scientifics, Crewe, UK), followed by centrifugation (18,000× *g*, 1 min). After mixing and centrifugation, 500 mL was transferred into a 5-mm NMR tube for spectral acquisition. High resolution [^1^H] NMR spectra were recorded on a 600-MHz Bruker Avance spectrometer fitted with a 5-mm TCI proton-optimized triple resonance NMR inverse cryoprobe and a 60-slot autosampler (Bruker, Billerica, MA, USA). Sample temperature was controlled at 300 K. Each spectrum consisted of 128 scans of 32,768 complex data points with a spectral width of 14 ppm (acquisition time 1.95 s). The noesypr1d presaturation sequence was used to suppress the residual water signal with low power selective irradiation at the water frequency during the recycle delay (D1 = 2 s) and mixing time (D8 = 0.15 s). A 90° pulse length of 8.8 ms was set for all samples. Spectra were transformed with a 0.3 Hz line broadening and zero filling, manually phased, baseline corrected, and referenced by setting the trimethylsilylpropanoic acid methyl signal to 0 ppm. Details on parameter settings for 2-dimensional NMR can be found at Le Gall et al. [21].

### 2.6. Statistical Analysis

All other data analysis was performed in GraphPad Prism version 8 (GraphPad Software, CA, USA). All data are presented as means (S.E.M), unless otherwise stated. After identifying outliers using the ROUT method (q = 1%), data were checked for normality/equal variances using the Shapiro–Wilk test. For analysis of dietary intervention an ANOVA, or Kruskal–Wallis test was used followed by Tukey or Dunns’s multiple comparison, depending on the normality of data. P values of less than 0.05 were considered statistically significant.

Statistical analysis of metabolomics data was carried out using Metaboanalyst 5.0 [22]. Data were normalized by sum, scaled by mean centering, and log-transformed. Univariate analysis was carried out by one way ANOVA, followed by Tukey HSD. Dendrogram and heatmaps were created with Spearman and Ward. Heatmaps show the significant metabolites based upon ANOVA results.

Correlation analysis between metabolomics data and microbiome data was conducted using M2IA [23]. Missing values were filtered if present in more than 80% of samples or the relative standard deviation was smaller than 30% [24]. Remaining missing data values were handled using random forest. Data were normalized using total sum scaling. A correlation analysis between bacterial genus and metabolite profile across LF and LF psyllium groups was made using Spearman’s rank-order correlation analysis [25].

## 3. Results

### 3.1. Addition of DFs Has Profound Morphological Implications, Independant of Body Weight Gain

Body weight increased over the 10-week intervention period (Figure 1A), but the addition of soluble DF had no significant influence on body weight gain nor food intake (*p* < 0.05 Figure 1B,C). Despite this, internal morphological changes were evident, with small intestinal length (*p* < 0.0003 Figure 2A) and total cecum weight (*p* < 0.0001 Figure 2B) significantly increased, by 1.2-fold and 2-fold, respectively, in response to DF addition. Interestingly, a post hoc analysis revealed that this increase was more prominent in psyllium containing diets. The liver:body weight ratio remained unchanged across experimental (Figure 2C). However, TAG accumulation was influenced by DF intervention (*p* < 0.05 Figure 2D), with combination diet resulting in significantly lower TAG levels, as assessed by post hoc analysis. This was consistent with a visual inspection of the liver tissue (H&E staining), which indicated reduced lipid droplet formation and macrophage infiltration across all DF groups (Figure 2E).

### 3.2. Psyllium and Inulin Result in Divergent Modulation of the Cecal Microbiota

The V3–V4 hypervariable region of the 16S rRNA gene was PCR amplified from the cecal content, in order to explore microbial response to inulin and psyllium dietary addition (LFIn, LFPsy and LFInPsy). Alpha diversity measured using Shannon index, an indicator of richness and evenness within samples, was not significantly different across experimental groups (*p* = 0.19 Figure 3A). Beta diversity, addressed using a combination of weighted unifrac distance analysis and PERMANOVA, highlighted a robust response to DF addition, with LF inulin and LF psyllium showing a clear separation from each other, and the LF diet (*p* < 0.001 Figure 3B). A comparison of relative abundances through Kruskal–Wallis analysis corrected for multiple comparisons identified several key changes at the phylum, namely in Actinobacteria, Proteobacteria, Verrucomicrobia, Firmicutes, and Bacteroidetes (Figure 3C). P values and FDR are provided (Table 1). Notably there was an increase in the Bacteroidetes:Firmicutes ratio in response to the general addition of soluble DFs (*p* < 0.05 Figure 3D); however, only psyllium containing diets achieved significance. At the genus level *Bifidobacterium* was increased in response to inulin (Figure 4A), whilst *Parasutterella* (Figure 4B) and *Akkermansia* (Figure 4C) increased, *Enterorhabdus* (Figure 4D), *Odoribacter* (Figure 4E) and *Lachnoclostridium* genus (Figure 4F) were significantly reduced in psyllium containing diets. *p* values and FDR are provided (Table 2).

### 3.3. DFs Have a Profound Impact on Cecal Metabolomic Profile

In addition to the microbial analysis, ^1^H-NMR metabolomic profiling was conducted on the same caecal contents to gain insights into the metabolomic environment. Consistent with the microbiota, principal component analysis (PCA) of the metabolome showed clear separations of each distinct diet (Figure 5A), with the combination diet sharing characteristics of each constituent. This was supported by hierarchical clustering using Spearman and Ward which resulted in the formation of 4 robust clusters representing each dietary intervention (Figure 5B). Interestingly, the dendrogram indicated that of all the diets, LF psyllium was separated from LF by the greatest distance, therefore suggesting it to be the most distinct. This is apparent from the metabolite heatmap which highlights the 33 significantly altered metabolites (ANOVA *p* < 0.05) (Figure 5C). Many of these altered metabolites can be categorised as simple sugars, products of fermentation (e.g., SCFA’s); including butyrate, isovalerate, 2-methylbutyrate and proprionate (Figure 5D-G), TCA cycle intermediates or urea cycle metabolites. Since LF psyllium induced the greatest metabolomic change compared to LF, we performed a (Spearman) correlation between the microbiome and the metabolome data in order to elucidate potential interactions. Gut microbial profile correlates with azoles, organic acids and carbohydrates (Figure 5H) Further exploration of genera modulated by psyllium as determined in the aforementioned microbiome analysis, revealed that *Parasutterella* and *Akkermansia* had strong negative correlation with glucose (R −0.67, −0.73 respectively *p* < 0.05), type 2 diabetes associated metabolites 3-methyl-2-oxovaleric acid (R −0.72, −0.78 respectively *p* < 0.05), and citrulline (R −0.77, −0.71 respectively *p* < 0.05) [26,27], whilst *Enterorhabdus*, *Odoribacter* and *Lachnoclostridium* had the opposite effect (Table 3). Full correlation analysis table is given in Appendix A.

### 3.4. Soluble DF Addition Alters Intestinal Carbohydrate Response

With psyllium containing diets mitigating the obesogenic bacterial and metabolomic shift induced by the refined diet, we posited that psyllium may alter carbohydrate uptake. We therefore initially explored the intestinal gene expression profile of carbohydrate transporters and metabolising enzymes to determine the extent to which intestinal carbohydrate response was involved. Expression of the carbohydrate transporter genes *Slc2a2* and *Slc2a5* responsible for glucose and fructose absorption was significantly altered across the entirety of the intestine, with ileal, jejunal and particularly duodenal expression significantly altered (Figure 6A,B *p* < 0.05). Tukey or Dunn’s post-hoc analysis indicated that this was psyllium dependent with duodenal *Slc2a2* (Figure 6A) and *Slc2a5* (Figure 6B) expression reduced 56.74% and 57.67% respectively in response to LF psyllium (HED 18.3 g/day psyllium). Interestingly, inulin appeared to exert the opposite effect with *Slc2a2* and *Slc2a5* expression increasing distally. A similar response was noted when characterising the expression profile of carbohydrate metabolising enzymes. This was particularly true for the fructose metabolising enzymes *Khk* (Figure 6C) and *Fbp1* (Figure 6D). Although not as striking as the fructose related enzymatic profile, alterations in dietary starch metabolising enzymes, namely *Mgam* (dietary starch metabolism, Figure 6E) and *Sis* (dietary sucrose metabolism, Figure 6F), appeared to be generally reduced in response to psyllium and increased in response to inulin.

### 3.5. The Addition of Psyllium Leads to Altered Hepatic Carbohydrate Response and Lipogenesis

In order to assess how the aforementioned changes in intestinal carbohydrate response relate to liver health/function, hepatic carbohydrate metabolism and lipogenesis were assessed through gene expression analysis. Consistent with the intestinal transporter and metabolic enzyme profile, key carbohydrate metabolising enzymes were significantly modulated by dietary intervention, specifically the dietary fructose metabolising *Khk* (Figure 7A *p* < 0.01) and *Aldob* (Figure 7B *p* < 0.01) which were reduced 47.8% and 34.9% respectively by the LF psyllium intervention when compared to LF. Interestingly, in contrast to the intestinal profile this effect was also established in the combination diet. Carbohydrate sensing/lipogenesis promoting enzymes mirrored this metabolic profile with *Chrebp* (Figure 7C *p* = 0.093), *Irs1* (Figure 7D *p* < 0.05), *Srebf1* (Figure 7E *p* < 0.001), and *Fasn* (Figure 7F *p* < 0.01), downregulated 27.5%, 42.4%, 64.1% and 45.3%, respectively, in response to the addition of psyllium into the diet.

## 4. Discussion

There has been an increase in the consumption of refined foods depleted in certain micronutrients and essential bioactives, which if regularly consumed may trigger underlying health issues. Plant-food derived DFs are the components usually lacking from such diets and likely contribute to their deleterious nature. Given the well documented benefits associated with DF consumption [3], their use has been forwarded to mitigate the surge in nutritional diseases associated with a Western-style diet; however, the underlying mechanisms responsible for these effects are yet to be completely resolved. Furthermore, it is conceivable that distinctive DFs will have distinctive biological outcomes/actions [4]. Therefore, DFs should be investigated both individually, and in combination, to determine their strengths and limitations in the context of health and disease. Considering this, the present study examined the intestinal and hepatic carbohydrate response to a refined LF diet, categorizing the impact of soluble DF addition. Multi-omics approaches were employed to evaluate the involvement of the gut microbiota, providing a more complete overview of the microbiome–gut–liver axis as a whole.

The presence of soluble DFs is believed to increase post-meal satiety and decrease subsequent hunger, resulting in modest weight loss, attributable to lower energy density [28], although this effect is variable across the current literature [28,29,30]. In this study weight gain and caloric intake remained constant, regardless of soluble DF addition, although this may be a product of animal age, study duration, DF dosage, and or background diet. In stark contrast to body weight gain, internal morphological features, including intestinal length and cecum weight, were increased remarkably by soluble DF addition (particularly under a psyllium diet), indicative of intestinal epithelial cell and bacterial cell proliferation. The associated increase in intestinal length has been suggested to be microbiome-dependent, with the effect absent in germ-free mice [31]. Additionally, this has been purported to be mediated through SCFA production [32]. However, given the distinct differences observed between inulin and psyllium, at both the microbiome and metabolomics level, and the lack of SCFA upregulation by psyllium, one would speculate that greater complexity exists in the overall process. It has also been suggested, however, that psyllium leads to greater starch excretion and a shift in the fermentation site toward the distal colon, which may similarly explain this lack of SCFA accumulation in the cecum [33]. Follow up studies should, therefore, consider the microbiota across the entirety of the gastrointestinal tract for validation/confirmation.

Gene expression analysis across the entirety of the intestine revealed striking changes in the expression profile of carbohydrate transport and metabolising proteins in response to psyllium when compared to both LF and LF inulin. To our knowledge this is the first time that the expression profile of such markers has been categorised in response to psyllium and inulin addition. From the data one could speculate two potential modes of action: either inulin modulates metabolic upregulation and subsequently shields the liver from surplus carbohydrates [34], or psyllium inhibits/delays intestinal carbohydrate uptake, reducing carbohydrate availability. Evidence from gene expression analysis of intestinal and hepatic fructose metabolising enzymes as well as hepatic lipogenesis promoting genes supports the latter notion given the clear downregulation of these genes within the liver and across the duodenum, jejunum, ileum. However, both inulin and psyllium ameliorated the early signs of diet induced liver steatosis, as evident from the healthier (reduced lipid droplet/macrophage infiltration) livers when compared to the cellulose only diet, therefore indicating divergent but ultimately protective roles for both inulin and psyllium. This notion is furthered by the fact that TAG’s were reduced to a greater extent in the combination group, benefiting from both the inulin and psyllium dependent mechanisms. Based upon this evidence the psyllium dependent effects likely stem from its gel forming capability which increases the viscosity of bowel contents, therefore delaying and reducing luminal carbohydrate absorption [30,35,36]. Conversely, inulin acts through non-absorption related mechanisms, potentially linked to its higher SCFA production or its impact upon bile acid metabolism.

Although alpha diversity did not reach significance, phylum level modulation was apparent as evidenced by the Bacteroidetes: Firmicutes ratio. Beta diversity was robustly and significantly altered by dietary intervention, with separation of soluble DF classes evident, indicative of divergent psyllium and inulin gut microbial modulation. The combination diet plotted between the two interventions thus indicating shared psyllium and inulin features. Consistent with an increase of Actinobacteria at the phylum level, a sizable increase in *Bifidobacteria* was observed in the inulin dietary intervention. This is a well-established response to inulin [37], with fermentation of inulin providing a nutrient source for the bacteria, which generate SCFAs and was somewhat established but to a lesser extent in the combination diet (dilution effect). This *Bifidobacteria* increase was completely absent in LF psyllium, additionally SCFA levels were much lower in the LF psyllium group, supporting the notion that psyllium is less fermentable [38], and supporting the likelihood that SCFA independent mechanisms are responsible for psyllium’s effects (although shifting of the fermentation site may mask this effect). Excluding inulin’s influence upon *Bifidobacteria*, psyllium appeared to have a more robust impact upon gut microbial genera. This may relate to psyllium’s ability to trap water in the intestine increasing stool water, GI transit time and ultimately the colonic environment [39]. Firstly, *Akkermansia* was exclusively increased by psyllium containing diets. *Akkermansia* is strongly associated with the modulation of energy metabolism and glucose tolerance, potentially through increased thermogenesis [40]. As such *Akkermansia* has been consistently recognised for its role in mitigating metabolic disease risk [41,42]. *Akkermansia* increased in response to both psyllium and combination diets, providing a potential basis for the established hepatoprotective effects. The significance of which is elevated, given the absence of altered intestinal carbohydrate response to the combination diet. In addition to the increase in *Akkermansia*, we observed further psyllium related genera changes including a significant increase of *Parasutterella*. Considered a core microbiome member [43], *Parasutterella* may reduce liver inflammation, under the actions of Forkhead box O transcription factors [44]. Finally, an apparent reduction of the purported obesity promoting *Odoribacter*, *Lachnoclostridium* and *Enterorhabdus* [45,46,47,48,49,50], was similarly evident in response to the psyllium containing diets.

As briefly touched upon the metabolomics profile reflected gut microbial changes, with *Bifidobacteria* correlating with the increased SCFA fermentation products linked to inulin containing diets. Conversely glucose levels and T2D related metabolites were negatively associated with *Akkermansia* and *Parasutterella* whilst positively associated with *Odoribacter*, *Lachnoclostridium* and *Enterorhabdus* suggesting that psyllium may somewhat alter the bacterial environment to a more protective and potentially anti-obesogenic profile. Excluding glucose, the branched chain amino (BCAA) acid 3-methyl-2-oxovalerate has been identified as the strongest predictor of impaired fasting glucose [51]. BCAA dysregulation closely coincides with mitochondrial dysfunction (which generates 3-methyl-2-oxovalerate). Therefore, increasing abundance of bacterial species such as *Akkermansia* which promote mitochondrial biogenesis and function [52], may be key to mitigating the impact of refined diets. Whether this microbial modulation is causal or causative still remains to be fully determined; although the hepatoprotective effects of the combination diet, which failed to elicit changes in carbohydrate transporter profiles provides a substantial case for microbial involvement in the established hepatoprotection.

Together, we report that inulin increases *Bifidobacteria* leading to greater SCFA concentrations, whilst psyllium containing diets elicited a more extensive microbial change. Interestingly both diets somewhat ameliorated the pathophysiological liver phenotype associated with a purified diet, highlighting the potential of DF’s in mitigating its deleterious effects. Given the gene expression profile observed within the liver and intestine, we propose that psyllium reduces carbohydrate, but particularly fructose uptake in turn regulating simple carbohydrate entry into the liver via the portal vein, preventing liver steatosis. However, this does not account for the hepatoprotective effects of the combination diet, therefore stressing the potential importance of the gut microbiota shift in this process. It is therefore likely that psyllium has both microbiota dependent and independent effects. From a translational perspective fibre content incorporated into these diets represents 18.3 g/day human equivalent which we consider achievable based upon the UK average (17 g for women and 20 g for men) (British Nutrition Foundation). Although still significantly less than the 30 g/day recommended by the NHS. Perhaps, making a case for greater consideration of DF type in individuals that struggle to meet these recommendations.

## 5. Conclusions

In conclusion, despite evidence of hepatoprotection across all DF interventions, the psyllium containing diets possessed the greatest capacity to regulate carbohydrate uptake and metabolism; highlighting that not all DFs are equal. Interestingly, the combination of both diets studied (LF InPsy) appeared to confer greater hepatoprotection (reduced TAGs), potentially related to the acquisition/accumulation of both inulin- and psyllium-related microbial changes. As such, psyllium containing fiber combinations should be further evaluated, in order to determine the optimal composition for refined diet mitigation and metabolic disease prevention. Additionally, such findings require clinical confirmation, in order to validate this response in humans.

## Figures and Tables

**Figure 1 nutrients-13-04278-f001:**
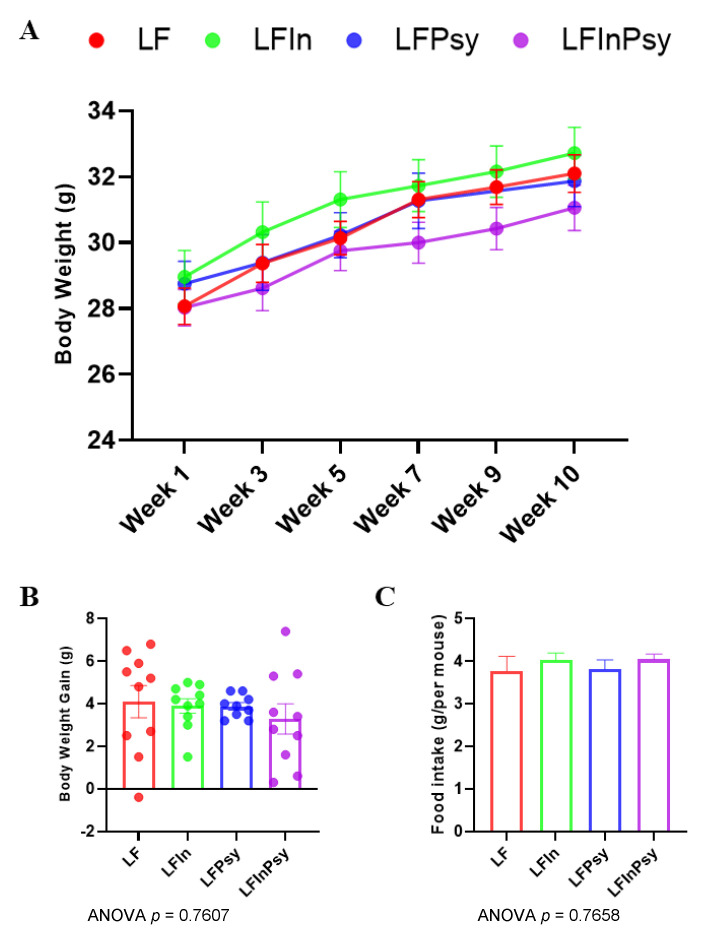
Soluble DF addition has no impact on body weight gain: (**A**) body weight trajectory across the 10-week study, (**B**) body weight gain, (**C**) and food intake remained constant across dietary interventions. Data are presented as mean ± S.E.M.

**Figure 2 nutrients-13-04278-f002:**
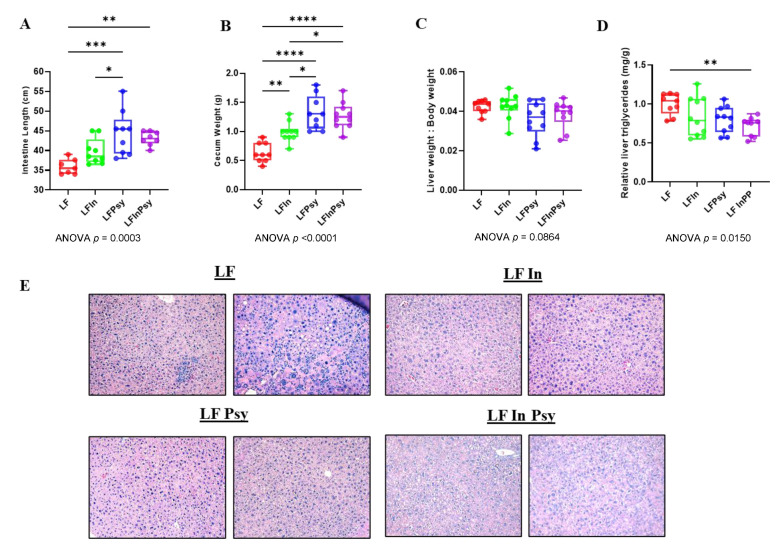
Addition of soluble DF to a refined LF animal diet alters gut–liver morphology (**A**) Intestinal length and (**B**) cecum weight were significantly increased in response to soluble DF addition, with the effect enhanced by psyllium containing diets. (**C**) Liver weight to body weight ratio was not significantly changed. (**D**) Hepatic TAGs were reduced by the combination diet, (**E**) consistent with the visually healthier liver tissue observed through H&E staining. Data are presented as mean ± S.E.M * *p* < 0.05, ** *p* < 0.01, *** *p* < 0.001, **** *p* < 0.0001.

**Figure 3 nutrients-13-04278-f003:**
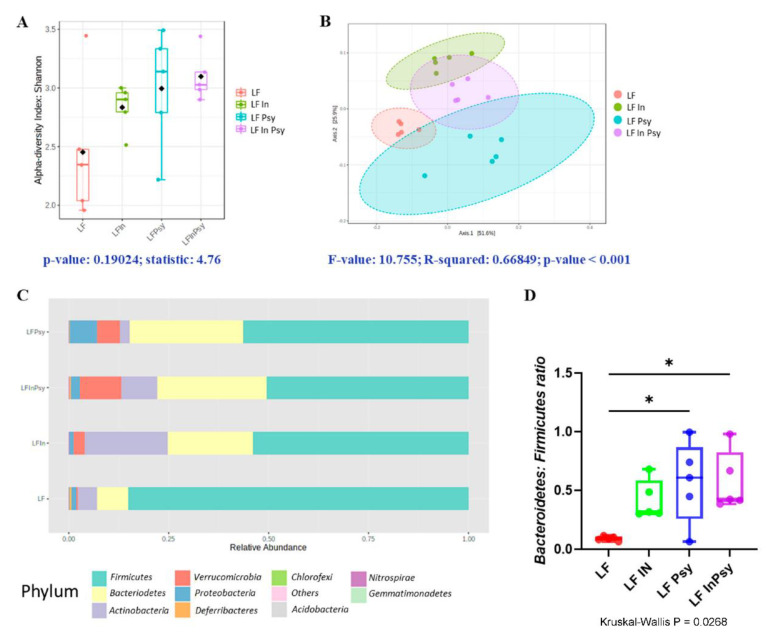
(**A**) α-diversity assessed by Shannon index was subtly but not significantly increased through DF addition, meanwhile (**B**) β-diversity, assessed using weighted unifrac analysis showed robust separation of DFs from the LF diet. (**C**) Graphical representation of phylum composition across each diet. (**D**) Bacteriodetes: Firmicutes ratio increased in response to psyllium containing diets (*p* values and FDR of significant changes are given in Table 1.). Data are presented as mean ± S.E.M * *p* < 0.05.

**Figure 4 nutrients-13-04278-f004:**
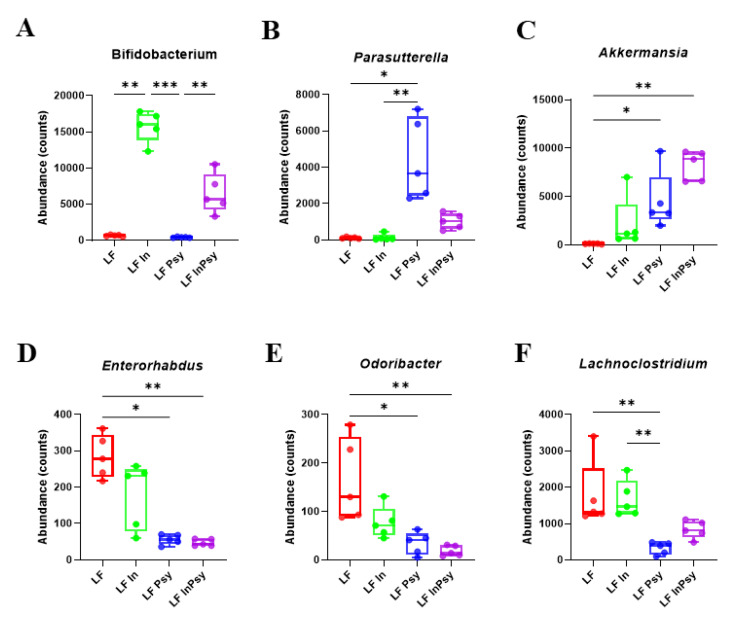
**(A**–**F**) Kruskal-Wallis univariate analysis revealed subsequent changes at the genus level with (**A**) *Bifidobacterium* increasing in response to inulin and (**B**) *Parasuterella* and (**C**) *Akkermansia* increasing in response to psyllium. Conversely (**D**) *Enterohabdus*, (**E**) *Odoribacter* and *Lachnoclostridium* reduced in response to psyllium addition. Data are presented as mean ± S.E.M * *p* < 0.05, ** *p* < 0.01, *** *p* < 0.001.

**Figure 5 nutrients-13-04278-f005:**
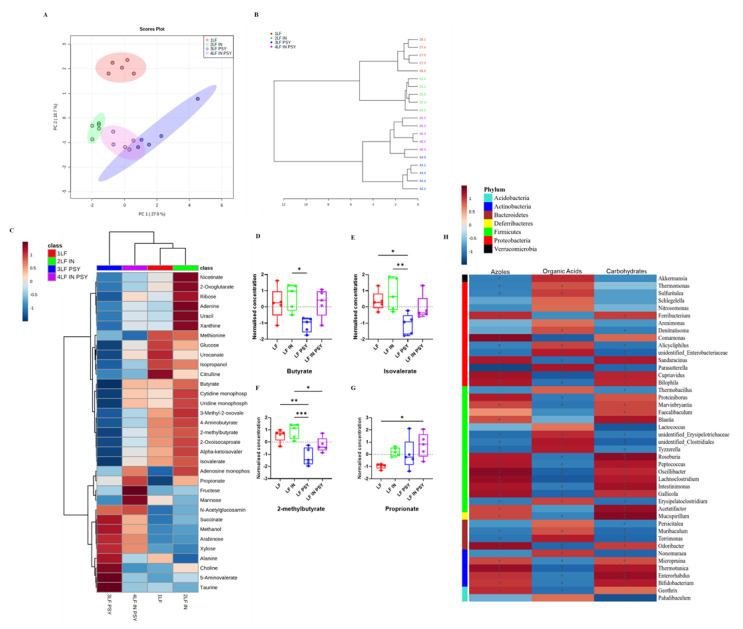
Metabolomic influence of DF and correlation with microbiota. (**A**) PCA plot shows a robust separation of the varying dietary groups, (**B**) consistent with the dendrogram, and (**C**) heatmap (significant metabolites), in which all four dietary groups formed four distinct clusters. (**D**–**G**) Concentration of significantly altered SCFA’s Butyrate, Isovalerate, 2-methylbutyrate, and propionate. (**H**) Interactions between the metabolome and microbiome of LF and LF psyllium groups were made using Spearman correlation analysis, which highlighted key changes (with asterisk) in Azoles, organic acids, and carbohydrates. Data are presented as mean ± S.E.M * *p* < 0.05, ** *p* < 0.01, *** *p* < 0.001.

**Figure 6 nutrients-13-04278-f006:**
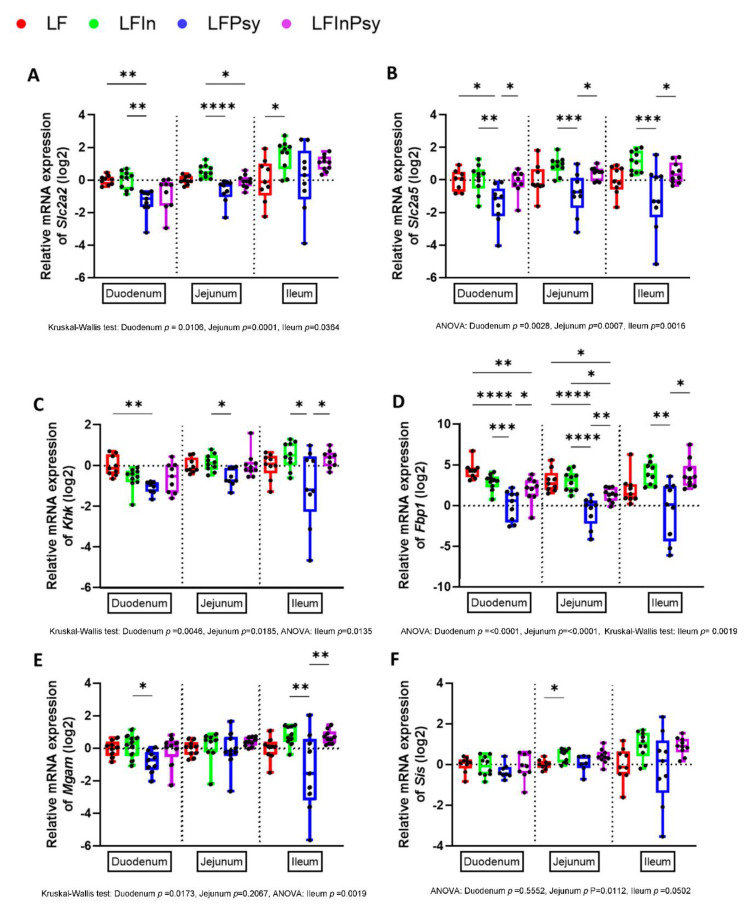
LF psyllium diet attenuates intestinal carbohydrate transport and metabolising potential: (**A**) Intestinal expression of carbohydrate transporter *Slc2a2* (**B**) and *Slc2a5* was reduced by psyllium intervention. (**C**) Carbohydrate metabolising enzymes *Khk*, (**D**) *Fbp1* (**E**) *Mgam*, and (**F**) *Sis* were similarly diminished by the actions of the LF psyllium intervention. Data are presented as mean ± S.E.M * *p* < 0.05, ** *p* < 0.01, *** *p* < 0.001, **** *p* < 0.0001.

**Figure 7 nutrients-13-04278-f007:**
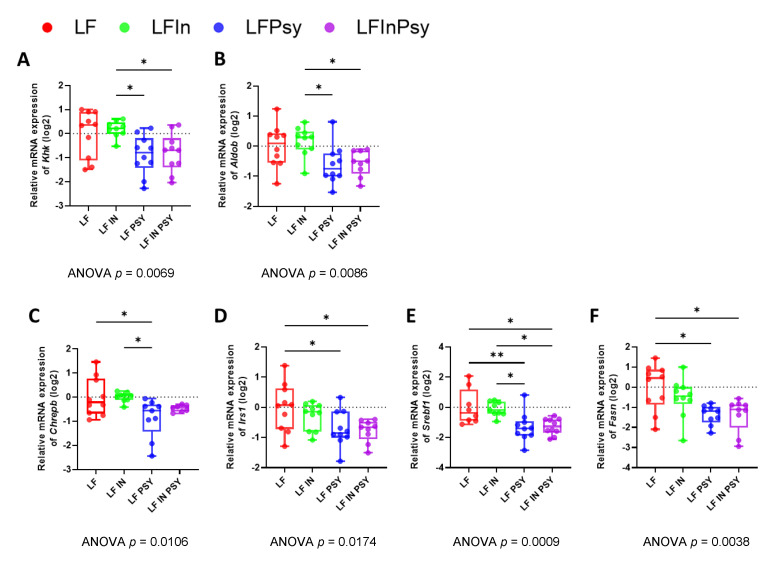
LF psyllium containing diets prevent fructose reaching the liver leading to subsequent reduction in lipogenesis: (**A**,**B**) Expression of fructose metabolising enzymes *Khk* and *Aldob* were reduced in the livers of psyllium supplemented animals. (**C**) Downregulation of *Chrebp* was indicative of reduced carbohydrate reaching the liver, (**D**–**F**) and in line with the reduction of lipogenesis promoting genes *Irs1*, *Srebf1* and *Fasn*. Data are presented as mean ± S.E.M * *p* < 0.05, ** *p* < 0.01.

**Table 1 nutrients-13-04278-t001:** *p* Values and FDR from classical univariate analysis at the phylum level.

Phylum	*p* Value	FDR
Actinobacteria	0.000892	0.008173
Proteobacteria	0.001026	0.008173
Verrucomicrobia	0.002157	0.009708
Deferribacteres	0.005	0.016724
Firmicutes	0.011603	0.029836

**Table 2 nutrients-13-04278-t002:** *p* Values and FDR from classical univariate analysis at genus level.

Genus	*p* Value	FDR
*Bifidobacterium*	0.00053615	0.035856
*Parasutterella*	0.00098285	0.035856
*Lachnoclostridium*	0.0012329	0.035856
*Odoribacter*	0.0019303	0.035856
*Akkermansia*	0.0021574	0.035856
*Enterorhabdus*	0.0023009	0.035856

**Table 3 nutrients-13-04278-t003:** A further exploration of the correlation analysis, focusing on genera of interest (those significant in microbiome analysis), indicated key metabolites that correlate with these bacterial genera.

Phylum.	Genus.	Metabolic Class	Metabolite	R	*p*-Value
Verrucomicrobia	*Akkermansia*	Amino Acids	Citrulline	−0.71	0.022
		Carbohydrates	Glucose	−0.733	0.016
		Others	3-Methyl-2-oxovalerate	−0.782	7.5 × 10^−^^3^
Proteobacteria	*Parasutterella*	Amino Acids	Citrulline	−0.77	9.2 × 10^−3^
		Carbohydrates	Glucose	−0.673	0.033
Actinobacteria	*Enterorhabdus*	Amino Acids	Citrulline	0.758	0.011
		Carbohydrates	Glucose	0.758	0.011
		Others	3-Methyl-2-oxovalerate	0.806	4.9 × 10^−3^
Proteobacteria	*Odoribacter*	Amino Acids	Citrulline	0.806	4.9 × 10^−3^
		Others	3-Methyl-2-oxovalerate	0.915	2 × 10^−4^
Firmicutes	*Lachnoclostridium*	Amino Acids	Citrulline	0.77	9.2 × 10^−3^
		Carbohydrates	Glucose	0.721	0.019
		Others	3-Methyl-2-oxovalerate	0.842	2.2 × 10^−3^

## Data Availability

Data will be made available upon request.

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
