# Peer review of "Differential Influence of Soluble Dietary Fibres on Intestinal and Hepatic Carbohydrate Response"

_nutrients, 2021, doi:10.3390/nu13124278_

Round 1

Reviewer 1 Report

Currently, the topics on the dietary fibres and multiomics approaches are of great interest to the scientific community and falls in the scope of this journal. I found this manuscript clear, detailed, well organized, and well written. Abstract adequately describes the study, principal results and conclusions. Data are properly analyzed and interpreted to support the conclusions. Tables and pictures are satisfactory and interpreted correctly.

I think that the conclusions are a bit short and should be extended.

There is only 1 tiny adjustment to make, the numbers must be added to the subparagraphs of point 2 (e.g., 2.1., 2.2., etc.).

Author Response

Point 1: Currently, the topics on the dietary fibres and multiomics approaches are of great interest to the scientific community and falls in the scope of this journal. I found this manuscript clear, detailed, well organized, and well written. Abstract adequately describes the study, principal results and conclusions. Data are properly analyzed and interpreted to support the conclusions. Tables and pictures are satisfactory and interpreted correctly.

Response 1: We thank the reviewer for their positive comments about our manuscript

Point 2: I think that the conclusions are a bit short and should be extended.

Response 2: We have extended conclusion section as requested. It now reads ‘In conclusion, despite evidence of hepatoprotection across all DF interventions, psyllium containing diets possessed the greater capacity to regulate carbohydrate uptake and metabolism, highlighting that not all DF’s are equal. Interestingly, the combination of both diets studied (LF InPsy) appeared to confer greater hepatoprotection (reduced TAGs), potentially relating to the acquisition/accumulation of both inulin and psyllium related microbial changes. As such, psyllium containing fibre combinations should be further evaluated in order to determine optimal composition for refined diet mitigation and metabolic disease prevention. Additionally, such findings require clinical confirmation in order to validate this response in humans.’

Point 3: There is only 1 tiny adjustment to make, the numbers must be added to the subparagraphs of point 2 (e.g., 2.1., 2.2., etc.).

Response 3: We thank reviewer 1 for this observation and we have now amended accordingly.

Reviewer 2 Report

Comments to authors :

The present works aims at establishing the role (beneficial) of soluble detary fibres on energy metabolism in the gut and liver. The present work is well described/designed, appropriately introduced and gives interesting and valuable data in the field of the gut-liver axis.

Nevertheless, I have several points that would require clarification:

Major comments :

Introduction part : can you justify the choice of the 2 soluble fibres chosen ? And associated to this, as you discuss a lot the metabolites potentially produced by the 2 fibres, can you discuss what can be expected in terms of fermentation profile concerning the 2 fibers. Indeed, some papers exist in the field and show that the fermentation profile of inulin and psyllium can be different (simply because the composition of both fibres in terms of oses are different: xylose and arabinose for psyllium – and fructose for inulin). To that respect, it is interesting to see that gut transporter for fructose are increased in ileum for inulin compared to Psy, and inulin is made of fructose and starts to be degraded in the distal part of the gut….). depending on this composition, what consequence on the fermentation profile ? Similarly, you comment some delay and alteration of transit and absorption site of other carbohydrates from the diet, can you make some assumption on potential impact of those 2 fibres on viscosity or the digesta for instance… that could alter transit time and alter digestibility of starch?

Statistics : I have difficulties in understanding some of the interpretations and hypothesis raised by the authors in the discussion for the following reason : The authors should be more careful in the extrapolation of numerical alterations they found. L 220-222, they comment alpha diversity and insist on the greater richness in fibre supplemented groups. Unfortunately, looking at the stats, there is no effect. Additionally, this potential increased diversity is commented elsewhere in the discussion. The authors should be more cautious when describing the data. For this parameter, no effect, not even a tendency. I would stick on that. But there are some other alterations at the phylum level, so the rest of the discussion can remain unaltered. Could you check, throughout the paper, that numerical non-significant effects are not detailed or commented ?

The authors decided to present the data on metabolomics using global effect of diets (PCA) and dendrogram. After this, they decided to only look at the correlation for microbiota composition and metabolomics between LF and LFPsy only.  This was justified by the fact that LFPsy was the group that was the most different from LF. This looks the case looking at the PCA but the other groups are also quite different from LF too…. Additionally, within the discussion part, there is quite some discussions on SCFAs (L347-353, but also elsewhere : L384-390….) and particularly on the absence of SCFA upregulation by psyllium (L349). Looking at the data presented in the paper, I have no way to validate this assumption or not. So, to make short, insert the data for at least cecal SCFAs in the 4 diets. This would strengthen the discussion part, I think.

Discussion part (and in line with the above comment): Basically, I would have appreciated, in order to be more precise and forceful on the gut-liver axis, to have a more detailed evaluation of the functionality of the microbiota from the animals fed the 4 different diets. The role of the SCFAs is hypothesized but no data shown in this field, whereas citrulline, glucose, 3-methyl-oxovalerate are discussed relatively to the associated bacteria (correlated + or -) but the discussion on their potential impact on gut and liver metabolism is very slight or non-existent. Fibres can have effects on both digestibility of the rest of carbohydrates from the diet (via some bulking effects of fibres). They can also have differential effect due to alteration of microbiota and the metabolites produced (and to this matter, mechanisms should be hypothesized depending on metabolites altered). These should appear more clearly in the discussion part.

It is mentioned that blood is withdrawn from the animals. Any measurements made in blood/plasma to evaluate the insulin sensitivity of the animals or other metabolic parameters ?

Lastly, you are using relative expression of genes relatively to LF. Why using this way of presenting data ? Generally, mRNA levels are presented as relative gene expression to the value of the housekeeping gene. And groups are compared subsequently.

Minor comments :

Basically, I find the way you present stats in the figures quite complex to understand (Fig 2B and 2D in particular)

Figure 2D: the stats seem to show that a significant difference is obtained between LF and LFinPsy but not between LF and other fibre supplemented groups. This is not consistent with what is described in the text.

Stats : you say that you are using ANOVA. Can you show the overall effect of your analysis and not only the post hoc stats ?

L347-349 : the increased caecum weight : is it full or empty ? So it is the tissue that adapted to an altered nutrients supply (i.e. fibers) or an increased caecum content ? If this is the latter, this only means that fibres reached the caecum, that increased the weight. By the way, I do not understand very well the stats for caecum weight. What the 1 and 2 stars on the same line refer to (Fig 2B)?

Figure 5:  could you add a title to the figure ? : difficult to know what data you are talking about.

In several figures : carbohydrates should be replaced by glucose, fructose, starch or an association of them, again to add clarity in the data you are describing.

L374-375 : If you do not want to show the data, I would sustain the assumption made by citation of the literature already existing on the subject and not on “not shown” data that cannot be verified.

Author Response

The present works aims at establishing the role (beneficial) of soluble detary fibres on energy metabolism in the gut and liver. The present work is well described/designed, appropriately introduced and gives interesting and valuable data in the field of the gut-liver axis.

General response: We are very pleased to receive such comments from reviewer 2 and hope that our responses can resolve any issues they may have. We have broken reviewer 2’s comments into smaller more specific issues for ease of understanding.

Point 1: Can you justify the choice of the 2 soluble fibres chosen?

Response 1: We thank the reviewer for their comment. We have now added the following sentence to the material and methods section L91 ‘ Inulin and psyllium were chosen as both popular commercially available supplementary soluble dietary fibers (particularly for mouse feed) with reported prebiotic/health implications. Additionally, the two fibers have been predominantly studied in isolation, but are rarely compared nor assessed in combination. Upon completion of the intervention, mice were sacrificed under terminal anaesthesia.’

Point 2: can you discuss what can be expected in terms of fermentation profile concerning the 2 fibres Indeed, some papers exist in the field and show that the fermentation profile of inulin and psyllium can be different (simply because the composition of both fibres in terms of oses are different: xylose and arabinose for psyllium – and fructose for inulin).

Response 2: Thank you for the valuable suggestion. The following paragraph has been added to the introduction L50: ‘The physiological benefits associated with fibre types relates to its chemical composition which in turn dictates fermentability and solubility (reviewed extensively by [8]). Briefly, inulin is purportedly highly fermentable whilst psyllium is only moderately fermentable. Fermentation in the distal small intestine and proximal colon provides energy and an array of metabolic substrates e.g. SCFA, which are believed to promote specific microbiota alterations, leading to different fermentation patterns. For example, inulin reportedly leads to the preferential growth of the bacteria Lactobacilli and Bifidobacteria. In contrast less fermentable, intermediate soluble fibres, such as psyllium have a higher water-holding / gel-forming capacity which may alter gastric emptying/nutrient absorption ultimately affecting glucose and lipid absorption

Point 3: What consequence on the fermentation profile? Similarly, you comment some delay and alteration of transit and absorption site of other carbohydrates from the diet, can you make some assumption on potential impact of those 2 fibres on viscosity or the digesta for instance… that could alter transit time and alter digestibility of starch?

Response 3: Please refer to response 2 which should also resolve this issue.

Point 4: L 220-222,. Unfortunately, looking at the stats, there is no effect. Additionally, this potential increased diversity is commented elsewhere in the discussion.

Response 4: We thank reviewer for pointing this out. We have amended the text of the results and discussion accordingly. Results: L230 ‘Alpha diversity measured using Shannon index an indicator of richness and evenness within samples; was not significantly different across experimental groups (P = .19 Figure 3A).’ Results: L412 ‘Although alpha diversity did not reach significance, phylum level modulation was apparent as evidenced by the Bacteroidetes: Firmicutes ratio.’

Point 5: Could you check, throughout the paper, that numerical non-significant effects are not detailed or commented?

Response 5: We thank the reviewer for their suggestion. We have checked throughout the paper and have amended accordingly. Liver weight: body weight ratio: Results: L211 ‘Liver:body weight ratio remained unchanged across experimental groups 1C).’ Figure: L223 ‘Liver weight to body weight ratio was not significantly changed’

Point 6: There is quite some discussions on SCFAs (L347-353, but also elsewhere : L384-390….) and particularly on the absence of SCFA upregulation by psyllium (L349). Looking at the data presented in the paper, I have no way to validate this assumption or not. So, to make short, insert the data for at least cecal SCFAs in the 4 diets. This would strengthen the discussion part, I think.

Response 6: We thank reviewer for highlighting this issue. Levels of SCFAs were provided in the heatmap (figure 5C). However for clarity we now added independent graphs for each significant SCFA identified (Figure 5D-G).

Point 7:  More detailed evaluation of the functionality of the microbiota

Response 7: We would like to direct reviewer 2 towards the detailed discussion provided from L428 onwards in which the importance of the upregulated microbiota genera Akkermansia and Parasuterlla is described.

Pont 8: The role of the SCFAs is hypothesized but no data shown in this field

Response 8: See response 6 for detail of the SCFAs

Point 9: citrulline, glucose, 3-methyl-oxovalerate are discussed relatively to the associated bacteria (correlated + or -) but the discussion on their potential impact on gut and liver metabolism is very slight or non-existent. Fibres can have effects on both digestibility of the rest of carbohydrates from the diet (via some bulking effects of fibres). They can also have differential effect due to alteration of microbiota and the metabolites produced (and to this matter, mechanisms should be hypothesized depending on metabolites altered). These should appear more clearly in the discussion part.

Response 9: We thank the reviewer for their suggestion. We have added further mechanistic hypotheses: L448 ‘Excluding glucose, the branched chain amino (BCAA) acid 3-methyl-2-oxovalerate has been identified as the strongest predictor of impaired fasting glucose [51]. BCAA dysregulation closely coincides with mitochondrial dysfunction (which generates 3-methyl-2-oxovalerate). Therefore, increasing abundance of bacterial species such as Akkermansia which promote mitochondrial biogenesis and function [52], may be key to mitigating the impact of refined diets.

L418 ‘This is a well-established response to inulin [37], with fermentation of inulin providing a nutrient source for the bacteria, which generate SCFAs and was somewhat established but to a lesser extent in the combination diet (dilution effect)

L427 ‘This may relate to psyllium’s ability to trap water in the intestine increasing stool water, GI transit time and ultimately the colonic environment [39].

Point 10: It is mentioned that blood is withdrawn from the animals. Any measurements made in blood/plasma to evaluate the insulin sensitivity of the animals or other metabolic parameters?

Response 10: We agree that this information would have been interesting, unfortunately this was an error in our methodology section and blood was not withdrawn.  

Point 11: Lastly, you are using relative expression of genes relatively to LF. Why using this way of presenting data ? Generally, mRNA levels are presented as relative gene expression to the value of the housekeeping gene. And groups are compared subsequently.

Response 11: We thank the reviewer comments. We employed the ΔΔCT method which determines gene expression relative to housekeeping gene levels (here normalised against Tbp). Our data are then normalised compared to LF. We have amended the results section as follows: ‘Results are expressed as relative quantity scaled to the average across all samples per target gene and normalized to the reference gene, results are presented as log2 fold change.’

Point 12: Basically, I find the way you present stats in the figures quite complex to understand (Fig 2B and 2D in particular) the stats seem to show that a significant difference is obtained between LF and LFinPsy but not between LF and other fibre supplemented groups. This is not consistent with what is described in the text.

Response 12: We believe some confusion may have arose from the software used for making graphs which connected some of the statistical comparisons on figure 2B, this has now been resolved. As for figure 2D this was correct. We mentioned a nominal decrease across all fibres not significant. However, we have now removed this from the text. L212 ‘However, TAG accumulation was influenced by DF intervention (P < .05 Figure 2D), with combination diet resulting in significantly lower TAG levels    as assessed by post-hoc analysis.’

Point 13: Stats : you say that you are using ANOVA. Can you show the overall effect of your analysis and not only the post hoc stats?

Response 13: Overall effect has been now added. It can be found either in tables (microbiota data), or is underneath respective figures.

Point 14: L347-349 : the increased caecum weight : is it full or empty ? So it is the tissue that adapted to an altered nutrients supply (i.e. fibers) or an increased caecum content? If this is the latter, this only means that fibres reached the caecum, that increased the weight. By the way, I do not understand very well the stats for caecum weight. What the 1 and 2 stars on the same line refer to (Fig 2B)?

Response 14: Fig 2B was addressed in the previous comment Point 12. The cecum weight is indeed relating to content.

Point 15: Figure 5:  could you add a title to the figure ? : difficult to know what data you are talking about.

Response 15: A title was added L287 ‘Metabolomic influence of DF and correlation a with the gut microbiota’

Point 16: In several figures : carbohydrates should be replaced by glucose, fructose, starch or an association of them, again to add clarity in the data you are describing.

Response 16: We understand that this may have caused some confusion. We have now removed from the figure. The text and figure legends describe which molecules e.g. fructose or glucose are targeted specifically.

Point 17: L374-375 : If you do not want to show the data, I would sustain the assumption made by citation of the literature already existing on the subject and not on “not shown” data that cannot be verified.

Response 17: We have removed this from the manuscript
